# Novel First-Trimester Prediction Model for Any Type of Preterm Birth Occurring before 37 Gestational Weeks in the Absence of Other Pregnancy-Related Complications Based on Cardiovascular Disease-Associated MicroRNAs and Basic Maternal Clinical Characteristics

**DOI:** 10.3390/biomedicines10102591

**Published:** 2022-10-15

**Authors:** Ilona Hromadnikova, Katerina Kotlabova, Ladislav Krofta

**Affiliations:** 1Department of Molecular Biology and Cell Pathology, Third Faculty of Medicine, Charles University, 10000 Prague, Czech Republic; 2Institute for the Care of the Mother and Child, Third Faculty of Medicine, Charles University, 14700 Prague, Czech Republic

**Keywords:** cardiovascular microRNAs, early gestation, expression, prediction, preterm delivery, preterm prelabor rupture of membranes, screening, spontaneous preterm birth, peripheral venous blood

## Abstract

The goal of the study was to establish an efficient first-trimester predictive model for any type of preterm birth before 37 gestational weeks (spontaneous preterm birth (PTB) or preterm prelabor rupture of membranes (PPROM)) in the absence of other pregnancy-related complications, such as gestational hypertension, preeclampsia, fetal growth restriction, or small for gestational age. The retrospective study was performed in the period from 11/2012 to 3/2020. Peripheral blood samples were collected from 6440 Caucasian individuals involving 41 PTB and 65 PPROM singleton pregnancies. A control group with 80 singleton term pregnancies was selected on the basis of equal sample-storage time. A combination of only six microRNAs (miR-16-5p, miR-21-5p, miR-24-3p, miR-133a-3p, miR-155-5p, and miR-210-3p; AUC 0.812, *p* < 0.001, 70.75% sensitivity, 78.75% specificity, cut-off > 0.652) could predict preterm delivery before 37 gestational weeks in early stages of gestation in 52.83% of pregnancies with a 10.0% FPR. This predictive model for preterm birth based on aberrant microRNA expression profile was further improved via implementation of maternal clinical characteristics (maternal age and BMI at early stages of gestation, infertility treatment with assisted reproductive technology, occurrence of preterm delivery before 37 gestational weeks in previous pregnancy(ies), and presence of any kind of autoimmune disease (rheumatoid arthritis, systemic lupus erythematosus, antiphospholipid syndrome, type 1 diabetes mellitus, or other autoimmune disease)). With this model, 69.81% of pregnancies destined to deliver before 37 gestational weeks were identified with a 10.0% FPR at early stages of gestation. When other clinical variables as well as those mentioned above—such as positive first-trimester screening for early preeclampsia with onset before 34 gestational weeks and/or fetal growth restriction with onset before 37 gestational weeks using the Fetal Medicine Foundation algorithm, as well as positive first-trimester screening for spontaneous preterm birth with onset before 34 gestational weeks using the Fetal Medicine Foundation algorithm—were added to the predictive model for preterm birth, the predictive power was even slightly increased to 71.70% with a 10.0% FPR. Nevertheless, we prefer to keep the first-trimester screening for any type of preterm birth occurring before 37 gestational weeks in the absence of other pregnancy-related complications as simple as possible.

## 1. Introduction

Spontaneous preterm birth (PTB) is characterized as delivery before the completion of 37 gestational weeks induced by regular uterine contractions along with cervical changes. Preterm prelabor rupture of membranes (PPROM) is defined as leakage of amniotic fluid that precedes the onset of labour by at least 2 h [1,2,3].

An algorithm for the prediction of PTB with onset before 34 gestational weeks in the early stages of gestation exists that is based only on maternal characteristics (maternal age and BMI at early stages of gestation, racial origin, method of conception, and smoking during pregnancy) combined with previous obstetrics history [4]. It can identify, with a 10.0% false-positive rate (FPR), PTBs occurring before 34 gestational weeks in approximately 20% of nulliparous pregnancies and 38% of multiparous pregnancies [4]. This algorithm has been integrated into routine first-trimester prenatal screening performed within 11 and 13 gestational weeks and is currently used by the majority of Fetal Medicine Centres to calculate not only the risks for chromosomal aneuploidies (trisomy 21, trisomy 18, and trisomy 13) but also the risks for early preeclampsia (PE) with onset before 34 gestational weeks and fetal growth restriction (FGR) with onset before 37 gestational weeks (Astraia Obstetrics program) [5,6,7,8,9].

Recently, we demonstrated that a set of 12 microRNA biomarkers can predict any type of preterm delivery that is uncomplicated by hypertensive and/or growth-restricted pregnancy disorders. This set of microRNA biomarkers (miR-16-5p, miR-20b-5p, miR-21-5p, miR-24-3p, miR-26a-5p, miR-92a-3p, miR-133a-3p, miR-145-5p, miR-146a-5p, miR-155-5p, miR-210-3p, and miR-342-3p) revealed 52.83% of pregnancies destined to deliver before 37 gestational weeks with a 10.0% FPR in early stages of gestation [10].

Similarly, we have shown that a combination of a minimal number of microRNAs (miR-16-5p, miR-21-5p, miR-24-3p, miR-133a-3p, miR-155-5p, and miR-210-3p) can also predict any type of preterm delivery before 37 gestational weeks in the absence of other pregnancy-related complications in early stages of gestation with similar discrimination power (AUC 0.812, *p* < 0.001, 70.75% sensitivity, 78.75% specificity, cut-off > 0.652, 52.83% of cases with a 10.0% FPR) [10].

The goal of this study was to establish an efficient first-trimester predictive model for any type of preterm birth before 37 gestational weeks (PTB or PPROM) in the absence of hypertensive and/or growth-restricted pregnancy disorders or gestational diabetes mellitus. Since other pregnancy-related complications have already been reported to be associated with an altered expression profile for cardiovascular disease-associated microRNAs by our research group [11,12,13], for this study, we only selected pregnancies with preterm delivery in the absence of other pregnancy-related complications. This strategy was implemented to see how specifically individual pregnancy-related complications altered the expression profile of cardiovascular disease-associated microRNAs in maternal whole peripheral blood.

We were interested in whether the predictive model for any type of preterm birth before 37 gestational weeks (PPROM or PTB) in the absence of other pregnancy-related complications based on an aberrant microRNA expression profile could be further improved via implementation of other maternal clinical characteristics.

## 2. Materials and Methods

### 2.1. Patient Cohort

This study was performed in a retrospective manner for the period from 11/2012 to 3/2020. The peripheral blood was sampled from 6440 singleton Caucasian pregnancies during the first-trimester prenatal screening. A total of 4469 pregnancies had complete medical records. Ultimately, 41 pregnancies were diagnosed as PTBs, and 65 pregnancies were confirmed to have diagnoses of PPROM, among which 29 preterm births occurred before 34 gestational weeks and 77 after 34 gestational weeks. Only those pregnancies that occurred in the absence of other pregnancy-related complications (GH, PE, FGR, SGA, or GDM) were involved in the study.

A control group (*n* = 80) consisted of normal-term pregnancies and was selected with regard to the equality of gestational age at the time of peripheral blood sampling and the equality of biological sample-storage time.

An informed consent form was signed by all participants involved in the study.

### 2.2. Combined First-Trimester Risk Analysis

A first-trimester gestation algorithm for the calculation of risks of chromosomal aneuploidies (trisomy 21, trisomy 18, and trisomy 13) and of early PE before 34 gestational weeks, FGR before 37 gestational weeks, and PTB before 34 gestational weeks was produced by Astraia Software gmbh, Germany (Astraia Obstetrics Programme), in close collaboration with the Fetal Medicine Foundation (FMF) [9].

Ten women from the group with preterm pregnancies (*n* = 106) were predicted to develop early PE before 34 gestational weeks and/or FGR before 37 gestational weeks. In compliance with the ACOG 2018 guidelines [14] and NICE 2019 guidelines [15], low-dose aspirin (ASA, 100 mg) was administered daily in the evening to seven of these ten pregnant women to decrease the risk of onset of early PE and/or FGR before 37 gestational weeks. ASA administration was started in the 13th gestational week at the latest. None of these women ultimately developed PE or FGR. In our control group, none of the women were predicted to develop early PE before 34 gestational weeks and/or FGR before 37 gestational weeks and, therefore, none received ASA.

Furthermore, 27 women (25.47% cases) from the group with preterm pregnancies (*n* = 106) were predicted to be at risk of PTB before 34 gestational weeks. The risk of PTB was below the cut-off value of 1:100. Four of these twenty-seven women were simultaneously found to be positive for early PE before 34 gestational weeks and/or FGR before 37 gestational weeks in first-trimester screening. ASA was ultimately given to two of these four women. Simultaneously, 6 out of 80 control term-pregnancies (7.50%) were also identified to be at risk of PTB before 34 gestational weeks.

### 2.3. Processing of Samples

Briefly, total RNA enriched for small RNAs was isolated from whole peripheral venous blood using an mirVana microRNA Isolation Kit (Ambion, Austin, TX, USA). Reverse transcription was performed with a total reaction volume of 10 µL using miRNA-specific stem loop primers and a TaqMan MicroRNA Reverse Transcription Kit (Applied Biosystems, Branchburg, NJ, USA) [16,17,18,19]. Real-time qPCR reactions were performed with a total reaction volume of 15 µL consisting of 3 µL of cDNA, specific primers, TaqMan MGB probes, and the TaqMan Universal PCR Master Mix (Applied Biosystems, Branchburg, NJ, USA) in a 7500 Real-Time PCR System under standard TaqMan PCR conditions [16,17,18,19].

MicroRNA gene expression was assessed using the delta-delta Ct method as previously described [16,17,18,19,20]. MicroRNA gene expression data were normalized to the geometric mean of the Ct values of RNU58A and RNU38B, previously selected and tested endogenous controls [21].

### 2.4. Statistical Analysis

Initially, power analysis was performed to calculate the minimum sample size required (G^∗^Power Version 3.1.9.6, Franz Faul, University of Kiel, Germany). To achieve a power of 0.805, 51 cases and 51 controls needed to be tested. Similarly, to achieve a power of 0.902, 70 cases and 70 controls needed to be tested.

Unpaired nonparametric tests were used for statistical analyses (the Mann–Whitney test and Kruskal–Wallis one-way analysis of variance with post hoc test for comparison between groups). Afterwards, Benjamini–Hochberg correction was applied.

Receiver operating characteristic (ROC) curve analysis was performed for each individual microRNA biomarker to assess the area under the curve (AUC), the cut-off point-associated sensitivity, specificity, positive and negative likelihood ratios (LR+, LR-), and sensitivity at 90.0% specificity (10.0% false-positive rate (FPR)) (MedCalc Software bvba, Ostend, Belgium).

To select microRNA combinations, logistic regression and ROC curve analyses were applied (MedCalc Software bvba, Ostend, Belgium).

To assess the detection rate of pregnancies with preterm birth before 37 gestational weeks on the base of a combination of microRNA biomarkers dysregulated in early stages of gestation and maternal clinical characteristics, logistic regression with subsequent ROC curve analyses were again applied (MedCalc Software bvba, Ostend, Belgium).

Concerning the microRNA expression studies only, both pilot and validation studies with two independent cohorts of patients were initially performed.

Then, experimental microRNA expression data obtained from a larger independent cohort of patients with histories of PPROM (*n* = 65) and PTB (*n* = 41) were reported [10]. The predictive model for any type of preterm birth before 37 gestational weeks in the absence of other pregnancy-related complications was based on the aberrant microRNA expression profile and clinical characteristics of these patients with histories of PPROM (*n* = 65) and PTB (*n* = 41). Model validation studies would have been beneficial and desirable but would have required further long-term collection of samples within the framework of routine first-trimester prenatal screening to achieve a sufficient number of cases.

## 3. Results

### 3.1. Clinical Characteristics of Preterm Birth and Control Pregnancies

The clinical characteristics of the preterm birth and control pregnancies are summarized in Table 1.

From the clinical characteristics of the patients, it is obvious that the occurrence of preterm delivery before 37 gestational weeks in previous pregnancy(ies), the presence of any kind of autoimmune disease (rheumatoid arthritis, systemic lupus erythematosus, antiphospholipid syndrome, type 1 diabetes mellitus, or other autoimmune disease), and positive first-trimester screening for early PE before 34 gestational weeks and/or FGR before 37 gestational weeks using the FMF algorithm, together with positive first-trimester screening for PTB before 34 gestational weeks, represent independent significant risk factors for the subsequent onset of any type of preterm birth before 37 gestational weeks (PTB or PPROM).

### 3.2. The First-Trimester Prediction Model for Preterm Birth before 37 Gestational Weeks Based on the Combination of Six MicroRNA Biomarkers and a Minimal Number of Maternal Clinical Characteristics

The combination of basic maternal clinical characteristics (age and body mass index values at first trimester of gestation, treatment of infertility by techniques of assisted reproduction, preterm birth < 37 gestational weeks in previous pregnancy(ies), and the presence of any kind of autoimmune disease (RA, SLE, APS, T1DM, or other autoimmunity)) and six dysregulated microRNA biomarkers (miR-16-5p, miR-21-5p, miR-24-3p, miR-133a-3p, miR-155-5p, and miR-210-3p) demonstrated very good accuracy for the identification of pregnancies destined to deliver before 37 gestational weeks (AUC 0.874, *p* < 0.001, 73.58% sensitivity, 88.75% specificity, cut-off > 0.642930). It revealed, in early stages of gestation, 69.81% of pregnancies destined to deliver before 37 gestational weeks with a 10.0% FPR (Figure 1). The predictive ability to identify pregnancies destined to deliver before 37 weeks of gestation using the five clinical characteristics only was also evaluated (Figure 2).

### 3.3. The First-Trimester Prediction Model for Preterm Birth before 37 Gestational Weeks Based on the Combination of 12 MicroRNA Biomarkers and a Minimal Number of Maternal Clinical Characteristics

The basic maternal clinical characteristics (age and body mass index values at first trimester of gestation, treatment of infertility by techniques of assisted reproduction, preterm birth < 37 gestational weeks in previous pregnancy(ies), and the presence of any kind of autoimmune disease (RA, SLE, APS, T1DM, or other autoimmunity)) combined with 12 dysregulated microRNA biomarkers (miR-16-5p, miR-20b-5p, miR-21-5p, miR-24-3p, miR-26a-5p, miR-92a-3p, miR-133a-3p, miR-145-5p, miR-146a-5p, miR-155-5p, miR-210-3p, and miR-342-3p) also demonstrated very good accuracy for the identification of pregnancies destined to deliver before 37 gestational weeks (AUC 0.877, *p* < 0.001, 75.47% sensitivity, 87.50% specificity, cut-off > 0.639464). It revealed, in early stages of gestation, 66.98% of pregnancies destined to deliver before 37 gestational weeks with a 10.0% FPR (Figure 3).

### 3.4. The First-Trimester Prediction Model for Preterm Birth before 37 Gestational Weeks Based on the Combination of Six MicroRNA Biomarkers and the Maximal Number of Maternal Clinical Characteristics

The combination of seven maternal clinical characteristics (age and body mass index values at first trimester of gestation, treatment of infertility by techniques of assisted reproduction, preterm birth < 37 gestational weeks in previous pregnancy(ies), the presence of any kind of autoimmune disease (RA, SLE, APS, T1DM, or other autoimmunity), positive first-trimester screening for early PE before 34 gestational weeks and/or FGR before 37 gestational weeks using the FMF algorithm, and positive first-trimester screening for PTB before 34 weeks of gestation using the FMF algorithm) with six dysregulated microRNA biomarkers (miR-16-5p, miR-21-5p, miR-24-3p, miR-133a-3p, miR-155-5p, and miR-210-3p) showed the highest possible accuracy for the early identification of pregnancies destined to deliver before 37 gestational weeks (AUC 0.879, *p* < 0.001, 71.70% sensitivity, 91.25% specificity, cut-off > 0.609772). It revealed, in early stages of gestation, 71.70% pregnancies destined to deliver before 37 gestational weeks with a 10.0% FPR (Figure 4). The predictive ability to identify pregnancies destined to deliver before 37 weeks of gestation using the seven clinical characteristics only was also evaluated (Figure 5).

### 3.5. The First-Trimester Prediction Model for Preterm Birth before 37 Gestational Weeks Based on the Combination of 12 MicroRNA Biomarkers and the Maximal Number of Maternal Clinical Characteristics

The combination of seven maternal clinical characteristics (age and body mass index values at first trimester of gestation, treatment of infertility with techniques of assisted reproduction, preterm birth < 37 gestational weeks in previous pregnancy(ies), the presence of any kind of autoimmune disease (RA, SLE, APS, T1DM, or other autoimmunity), positive first-trimester screening for early PE before 34 gestational weeks and/or FGR before 37 gestational weeks using the FMF algorithm, and positive first-trimester screening for PTB before 34 gestational weeks using the FMF algorithm) and 12 dysregulated microRNA biomarkers (miR-16-5p, miR-20b-5p, miR-21-5p, miR-24-3p, miR-26a-5p, miR-92a-3p, miR-133a-3p, miR-145-5p, miR-146a-5p, miR-155-5p, miR-210-3p, and miR-342-3p) also showed the highest possible accuracy for the early identification of pregnancies destined to deliver before 37 gestational weeks (AUC 0.887, *p* < 0.001, 73.58% sensitivity, 90.00% specificity, cut-off > 0.651956). It revealed, in early stages of gestation, 73.58% of pregnancies destined to deliver before 37 gestational weeks with a 10.0% FPR (Figure 6).

## 4. Discussion

The pathogenesis of preterm birth is very complex since it involves multiple pathogenic mechanisms, from premature aging of fetal membranes, cervical incompetence, and uterine anomalies to chorioamnionitis [3,22,23,24,25,26,27,28,29,30]. We have recently shown that microRNAs dysregulated in early stages of gestation in mothers at risk of preterm birth interact with specific genes involved in key biological pathways related to preterm birth, such as apoptosis, inflammatory response, senescence, and autophagy [10]. Furthermore, the expression profiles of series of genes playing roles in the pathogenesis of preterm birth were observed to be altered in maternal circulation (peripheral blood) during the onset of PTB within 27–36 gestational weeks [31]. A meta-analysis of transcriptomic data derived from the mother and the fetus revealed 210 genes with differing expression overall (65 upregulated genes and 145 downregulated genes) in maternal circulation (peripheral blood) in late gestation in PTB pregnancies, and substantial numbers of these genes were immune-related [31].

Maternal whole peripheral blood reflects the level of maternal stress and maternal inflammatory response to pathological events. De novo synthesis of proteins appears in maternal peripheral blood leukocytes when cells are exposed to stressful stimuli. This event is preceded by a decrease in appropriate microRNA levels, which obviates degradation or blockage of translation of newly synthetized messenger RNA (mRNA) targets.

It is also generally accepted that progesterone plays an important role in the maintenance of uterine quiescence until the perinatal period. Progesterone maintains uterine quiescence and reduces uterine contractility through the mediation of multiple actions involving membrane ion channels, expression of receptors and agonists, intracellular cAMP and protein kinase A (PKA) signaling, calcium signaling, and contractile proteins [32]. At the least, the individual components of the cAMP-PKA pathway [33] are directed by several microRNAs, which was a subject of our interest. The beginning of the loss of uterine quiescence may be accompanied by the downregulation of microRNAs during the early stages of gestation in pregnancies with subsequent preterm delivery [10]. For example, the expression of RIIα protein encoded by the PRKAR2A gene, a form of particulate type II PKA, is directed by microRNAs (miR-20b-5p, miR-24-3p, miR-26a-5p, miR-133a-3p, miR-145-5p, and miR-342-3p) observed to be downregulated during the first trimester of gestation in pregnancies with subsequent preterm delivery [10]. In addition, expression of A kinase-anchoring proteins (AKAPs), such as AKAP95 and AKAP79, interacting with myometrial RIIα subunits is also directed by microRNAs that were downregulated during the first trimester of gestation in pregnancies with subsequent preterm delivery [10]. AKAP95 gene expression is directed via miR-20b-5p, miR-21-5p, miR-24-3p, miR-145-5p, miR-146a-5p, and miR-210-3p. AKAP79 gene expression is directed via miR-20b-5p, miR-21-5p, miR-24-3p, miR-26a-5p, miR-92a-3p, miR-133a-3p, miR-145-5p, and miR-155-5p.

A set of 12 microRNA biomarkers alone (miR-16-5p, miR-20b-5p, miR-21-5p, miR-24-3p, miR-26a-5p, miR-92a-3p, miR-133a-3p, miR-145-5p, miR-146a-5p, miR-155-5p, miR-210-3p, and miR-342-3p) demonstrated similar discrimination power in predicting during early stages of gestation any type of preterm delivery (PTB or PPROM) occurring before 37 gestational weeks in the absence of hypertensive and/or growth-restricted pregnancy disorders or gestational diabetes mellitus as the set of six microRNA biomarkers alone (miR-16-5p, miR-21-5p, miR-24-3p, miR-133a-3p, miR-155-5p, and miR-210-3p). Both combinations of microRNAs alone were able to predict preterm delivery occurring before 37 gestational weeks in 52.83% of cases with a 10.0% FPR. Therefore, the combination of six microRNAs (miR-16-5p, miR-21-5p, miR-24-3p, miR-133a-3p, miR-155-5p, and miR-210-3p) was considered sufficient to for implementation as a fundamental variable in a model for the prediction in early stages of gestation of the occurrence of any type of preterm delivery (PTB or PPROM) occurring before 37 gestational weeks in the absence of other pregnancy-related complications, such as GH, PE, FGR, or SGA.

The implementation of other essential maternal clinical characteristics in this predictive model, such as maternal age and BMI at early stages of gestation, infertility treatment with assisted reproductive technology, the occurrence of preterm delivery before 37 gestational weeks in previous pregnancy(ies), and the presence of any kind of autoimmune disease, could significantly improve the detection rate of any type of preterm birth (PTB or PPROM) occurring before 37 gestational weeks in the absence of other pregnancy-related complications, such as GH, PE, FGR, or SGA. This model was able to identify 69.81% of cases with a 10.0% FPR at early stages of gestation.

The implementation of other variables in this predictive model for preterm birth, such as positive first-trimester screening for early PE with onset before 34 gestational weeks and/or FGR with onset before 37 gestational weeks and positive first-trimester screening for PTB with onset before 34 gestational weeks, both using the FMF algorithm, would be possible but is not necessary. It slightly increased the detection rate to 71.70% of cases with a 10.0% FPR.

## 5. Conclusions

The novel predictive model for any type of preterm birth (PTB or PPROM) occurring before 37 gestational weeks in the absence of other pregnancy-related complications, such as GH, PE, FGR, or SGA, was based on the combination of the expression profiles of six microRNA (miR-16-5p, miR-21-5p, miR-24-3p, miR-133a-3p, miR-155-5p, and miR-210-3p) in whole peripheral blood samples and basic maternal clinical characteristics, such as maternal age and BMI at early stages of gestation, infertility treatment with assisted reproductive technology, the occurrence of preterm delivery before 37 gestational weeks in previous pregnancy(ies), and the presence of any kind of autoimmune disease. It was able to predict 69.81% of cases in the early stages of gestation with a 10.0% FPR.

The implementation of additional variables in this predictive model, such as positive first-trimester screening for early PE with onset before 34 gestational weeks and/or FGR with onset before 37 gestational weeks and positive first-trimester screening for PTB with onset before 34 gestational weeks, both using the FMF algorithm, increased the detection rate of preterm births to 71.70% of cases with a 10.0% FPR.

Consecutive large-scale studies involving other ethnic groups and multiple pregnancies are needed to verify the data resulting from this pilot study. Future work will also involve the use of machine-learning techniques for predicting preterm labor, measurements of heart rate variability, and electrohysterography, a noninvasive procedure to monitor contractions during pregnancy.

## 6. Patents

National patent application—Industrial Property Office, Czech Republic (patent n. PV2021-562).

## Figures and Tables

**Figure 1 biomedicines-10-02591-f001:**
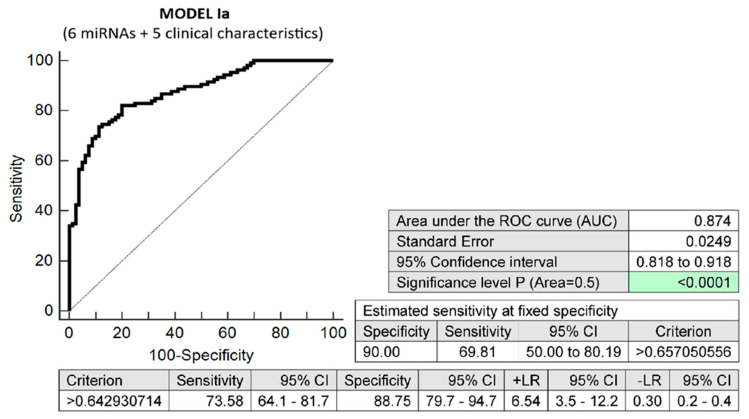
ROC analysis—prediction model Ia for preterm birth before 37 gestational weeks. Five clinical characteristics (age and body mass index values at first trimester of gestation, treatment of infertility by techniques of assisted reproduction, preterm birth < 37 gestational weeks in previous pregnancy(ies), and the presence of any kind of autoimmune disease (RA, SLE, APS, T1DM, or other autoimmunity)) combined with six microRNAs with aberrant expression profiles (miR-16-5p, miR-21-5p, miR-24-3p, miR-133a-3p, miR-155-5p, and miR-210-3p). In total, 69.81% of pregnancies destined to deliver before 37 weeks of gestation were identified during the first trimester of gestation with a 10.0% FPR.

**Figure 2 biomedicines-10-02591-f002:**
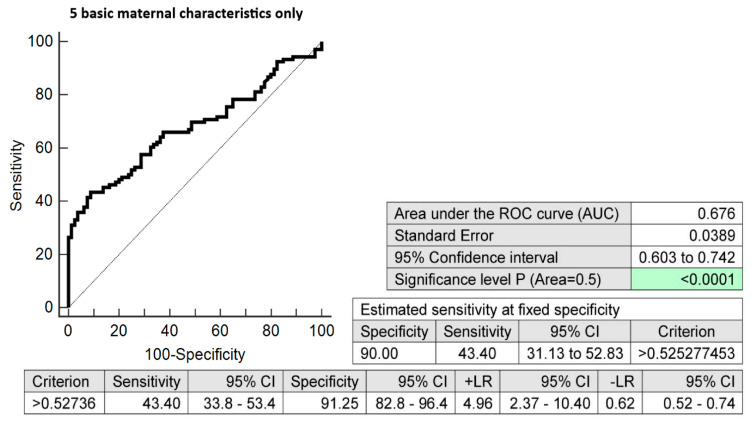
Combination of five basic maternal clinical characteristics only (maternal age and BMI values at early stages of gestation, infertility treatment with assisted reproductive technology, the occurrence of preterm delivery before 37 gestational weeks in previous pregnancy(ies), and the presence of any kind of autoimmune disease (rheumatoid arthritis, systemic lupus erythematosus, antiphospholipid syndrome, type 1 diabetes mellitus, or other autoimmune disease)). In total, 43.40% of pregnancies destined to deliver before 37 weeks of gestation were identified during the first trimester of gestation with a 10.0% FPR.

**Figure 3 biomedicines-10-02591-f003:**
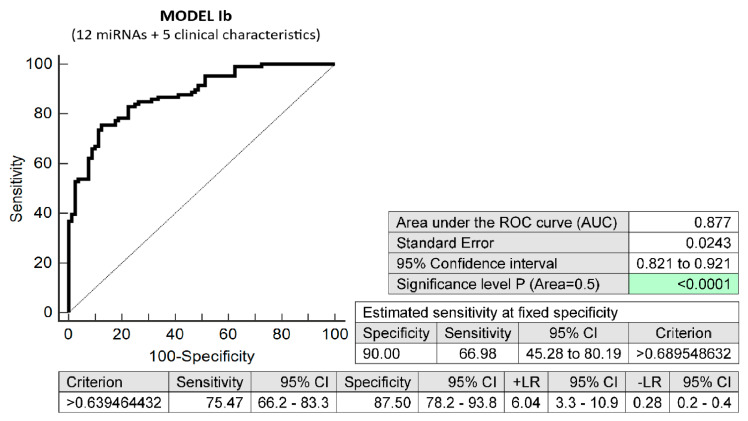
ROC analysis—prediction model Ib for preterm birth before 37 gestational weeks. Combination of five basic maternal clinical characteristics (age and body mass index values at first trimester of gestation, treatment of infertility with techniques of assisted reproduction, preterm birth < 37 gestational weeks in previous pregnancy(ies), and the presence of any kind of autoimmune disease (RA, SLE, APS, T1DM, or other autoimmunity)) and 12 dysregulated microRNA biomarkers (miR-16-5p, miR-20b-5p, miR-21-5p, miR-24-3p, miR-26a-5p, miR-92a-3p, miR-133a-3p, miR-145-5p, miR-146a-5p, miR-155-5p, miR-210-3p, and miR-342-3p). In total, 66.98% of pregnancies destined to deliver before 37 gestational weeks were identified during the first trimester of gestation with a 10.0% FPR.

**Figure 4 biomedicines-10-02591-f004:**
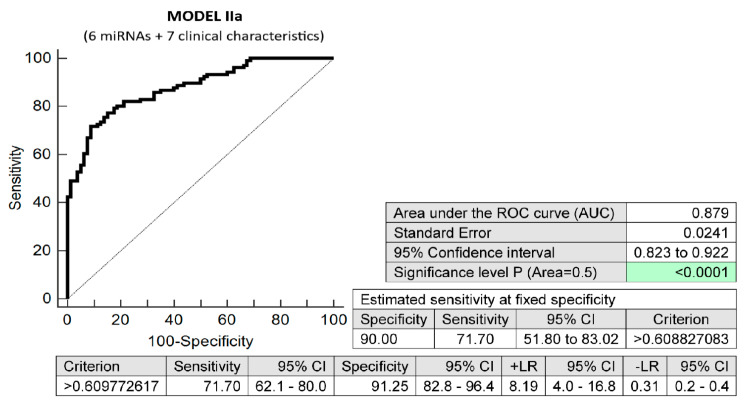
ROC analysis—prediction model IIa for preterm birth before 37 gestational weeks. Combination of seven maternal clinical characteristics (maternal age and BMI values at early stages of gestation, infertility treatment with assisted reproductive technology, the occurrence of preterm delivery before 37 gestational weeks in previous pregnancy(ies), the presence of any kind of autoimmune disease (rheumatoid arthritis, systemic lupus erythematosus, antiphospholipid syndrome, type 1 diabetes mellitus, or other autoimmune disease), positive first-trimester screening for early PE before 34 gestational weeks and/or FGR before 37 gestational weeks using FMF algorithm, and positive first-trimester screening for PTB before 34 gestational weeks using FMF algorithm) and six dysregulated microRNA biomarkers (miR-16-5p, miR-21-5p, miR-24-3p, miR-133a-3p, miR-155-5p, and miR-210-3p). In total, 71.70% of pregnancies destined to deliver before 37 gestational weeks were identified during the first trimester of gestation with a 10.0% FPR.

**Figure 5 biomedicines-10-02591-f005:**
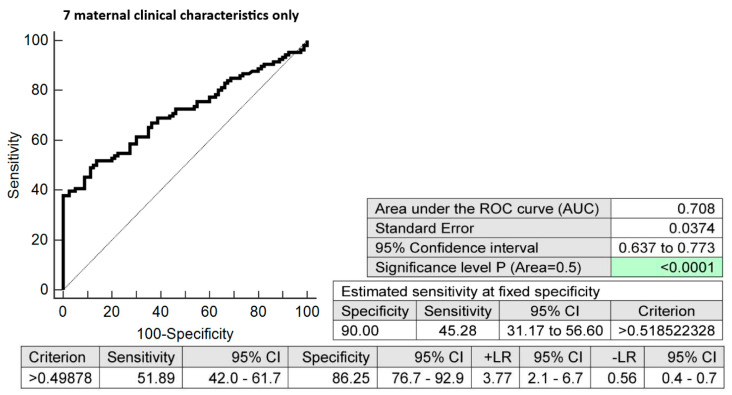
Combination of seven maternal clinical characteristics only (maternal age and BMI values at early stages of gestation, infertility treatment with assisted reproductive technology, the occurrence of preterm delivery before 37 gestational weeks in previous pregnancy(ies), the presence of any kind of autoimmune disease (rheumatoid arthritis, systemic lupus erythematosus, antiphospholipid syndrome, type 1 diabetes mellitus, or other autoimmune disease), positive first-trimester screening for early PE before 34 gestational weeks and/or FGR before 37 gestational weeks using FMF algorithm, and positive first-trimester screening for PTB before 34 gestational weeks using FMF algorithm). In total, 45.28% of pregnancies destined to deliver before 37 gestational weeks were identified during the first trimester of gestation with a 10.0% FPR.

**Figure 6 biomedicines-10-02591-f006:**
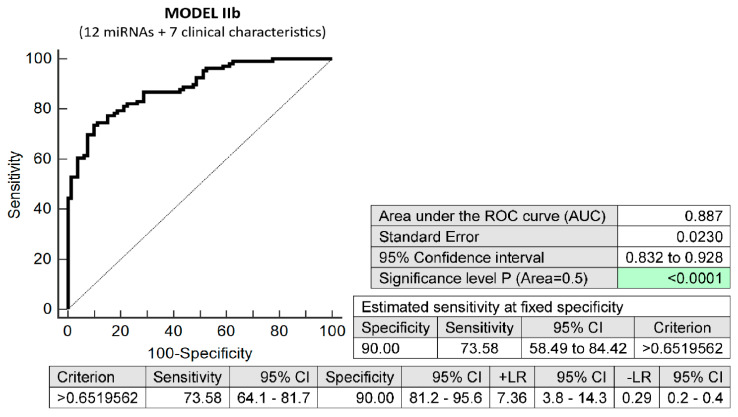
ROC analysis—prediction model IIb for preterm birth before 37 gestational weeks. Combination of seven maternal clinical characteristics (age and body mass index values at first trimester of gestation, treatment of infertility with techniques of assisted reproduction, preterm birth < 37 gestational weeks in previous pregnancy(ies), the presence of any kind of autoimmune disease (RA, SLE, APS, T1DM, or other autoimmunity), positive first-trimester screening for early PE before 34 gestational weeks and/or FGR before 37 gestational weeks using the FMF algorithm, and positive first-trimester screening for PTB before 34 gestational weeks using the FMF algorithm) and 12 dysregulated microRNA biomarkers (miR-16-5p, miR-20b-5p, miR-21-5p, miR-24-3p, miR-26a-5p, miR-92a-3p, miR-133a-3p, miR-145-5p, miR-146a-5p, miR-155-5p, miR-210-3p, and miR-342-3p). In total, 73.58% of pregnancies destined to deliver before 37 gestational weeks were identified during the first trimester of gestation with a 10.0% FPR.

**Table 1 biomedicines-10-02591-t001:** Clinical characteristics of the cases (preterm birth before 37 gestational weeks) and the controls.

	Normal Term Pregnancies (*n* = 80)	Preterm Birth (*n* = 106)	PTB (*n* = 41)	PPROM (*n* = 65)	*p*-Value ^1^ (95% CI)	*p*-Value ^2^ (95% CI)	*p*-Value ^3^ (95% CI)
* **Maternal characteristics** *							
Autoimmune diseases (SLE/APS/RA)	0 (0%)	2 (1.89%) 1 RA 1 SLE	0 (0%)	2 (3.08%) 1—RA 1—SLE	0.386 OR 3.852 (0.182–81.354)	0.742 OR 1.940 (0.038–99.526)	0.236 OR 6.339 (0.299–134.403)
Other autoimmune diseases	0 (0%)	5 (4.72%) 4—AIT 1—systemic scleroderma	2 (4.88%) 1—AIT 1—systemic scleroderma	3 (4.61%) 3—AIT	0.145 OR 8.724 (0.475–160.124)	0.137 OR 10.190 (0.478–217.368)	0.148 OR 9.016 (0.457–177.791)
Diabetes mellitus (T1DM)	0 (0%)	5 (4.72%)	3 (7.32%)	2 (3.08%)	0.145 OR 8.724 (0.475–160.124)	0.078 OR 14.636 (0.737–290.480)	0.236 OR 6.339 (0.299–134.403)
Diabetes mellitus (T2DM)	0 (0%)	0 (0%)	0 (0%)	0 (0%)	0.889 OR 0.756 (0.015–38.505)	0.742 OR 1.940 (0.038–99.526)	0.918 OR 1.229 (0.024–62.787)
Any kind of autoimmune disease (SLE/APS/RA/other/T1DM)	0 (0%)	12 (11.32%) 1—RA 1—SLE 1—SS 5—T1DM 4—AIT	5 (12.20%) 3—T1DM 1—AIT 1—SS	7 (10.77%) 1—RA 1—SLE 2—T1DM 3—AIT	0.035 OR 21.296 (1.241–365.357)	0.032 OR 24.260 (1.307–450.452)	0.040 OR 20.641 (1.156–368.625)
Trombophilic gene mutations	0 (0%)	6 (5.66%)	1 (2.44%)	5 (7.69%)	0.112 OR 10.413 (0.578–187.625)	0.278 OR 5.963 (0.238–149.670)	0.071 OR 14.636 (0.794–269.823)
Parity							
Nulliparous	41 (51.25%)	56 (52.83%)	14 (34.15%)	42 (64.61%)	0.831 OR 1.065 (0.596–1.905)	0.076 OR 0.493 (0.226–1.076)	0.107 OR 1.737 (0.888–3.399)
Parous—previous preterm delivery(ies) before 37 gestational weeks	0 (0%)	17 (16.04%)	11 (26.83%)	6 (9.23%)	0.017 OR 31.480 (1.863–531.988)	0.005 OR 60.705 (3.470–1062.101)	0.052 OR 17.588 (0.972–318.358)
Parous—previous term delivery(ies) after 37 gestational weeks	39 (48.75%)	33 (31.13%)	16 (39.02%)	17 (26.15%)	0.015 OR 0.475 (0.260–0.867)	0.310 OR 0.673 (0.313–1.447)	0.006 OR 0.372 (0.184–0.754)
History of miscarriage (spontaneous loss of a pregnancy before 22 weeks of gestation)	16 (20.0%)	26 (24.53%)	12 (29.27%)	14 (21.54%)	0.465 OR 1.300 (0.643–2.629)	0.255 OR 1.655 (0.695–3.941)	0.820 OR 1.098 (0.490–2.459)
History of perinatal death (the death of a baby between 22 weeks of gestation (or weighing 500 g) and 7 days after birth)	1 (1.25%)	3 (2.83%)	3 (7.32%)	0 (0%)	0.474 OR 2.301 (0.235–22.543)	0.118 OR 6.237 (0.628–61.962)	0.581 OR 0.405 (0.016–10.099)
ART (IVF/ICSI/other)	2 (2.5%)	8 (7.55%)	1 (2.44%)	7 (10.77%)	0.150 OR 3.184 (0.657–15.423)	0.984 OR 0.975 (0.086–11.081)	0.059 OR 4.707 (0.943–23.496)
Smoking during pregnancy	2 (2.5%)	5 (4.72%)	3 (7.32%)	2 (3.08%)	0.439 OR 1.931 (0.365–10.218)	0.229 OR 3.079 (0.493–19.208)	0.833 OR 1.238 (0.170–9.038)
* **Pregnancy details (first trimester of gestation)** *							
Maternal age (years)	32 (25–42)	32 (21–42)	33 (21–42)	32 (25–41)	0.353	0.706	1.0
Advanced maternal age (≥35 years old)	18 (22.5%)	32 (30.19%)	12 (29.27%)	20 (30.77%)	0.243 OR 1.489 (0.763–2.907)	0.416 OR 1.425 (0.607–3.345)	0.262 OR 1.531 (0.728–3.220)
BMI (kg/m^2^)	21.28 (17.16–29.76)	22.05 (16.51–33.5)	22.04 (17.96–31.83)	22.14 (16.51–33.5)	0.709	1.0	1.0
BMI ≥ 30 kg/m^2^	0 (0%)	6 (5.66%)	2 (4.88%)	4 (6.15%)	0.112 OR 10.413 (0.578–187.625)	0.137 OR 10.190 (0.478–217.368)	0.100 OR 11.780 (0.622–222.970)
Gestational age at sampling (weeks)	10.29 (9.57–13.71)	10.14 (9.43–14.57)	10.14 (9.43–12.86)	10.14 (9.86–14.57)	0.064	0.477	0.291
Screening—positive for spontaneous preterm birth (<34 weeks) with FMF algorithm	5 (6.25%)	25 (23.58%)	12 (29.27%)	13 (20.0%)	0.003 OR 4.630 (1.686–12.714)	0.002 OR 6.207 (2.009–19.174)	0.018 OR 3.750 (1.260–11.158)
Screening—positive for PE (<34 weeks) and/or FGR (<37 weeks) with FMF algorithm	0 (0%)	10 (9.43%)	2 (4.88%)	8 (12.31%)	0.049 OR 17.518 (1.011–303.599)	0.137 OR 10.190 (0.478–217.368)	0.031 OR 23.800 (1.346–420.697)
Aspirin intake during pregnancy	0 (0%)	7 (6.60%)	1 (2.44%)	6 (9.23%)	0.089 OR 12.136 (0.683–215.714)	0.278 OR 5.963 (0.238–149.670)	0.052 OR 17.588 (0.972–318.358)

Continuous variables, compared using the Mann–Whitney or Kruskal–Wallis tests, are presented as medians (range). Categorical variables, presented as numbers (percent), were compared using an odds ratio test. *p*-Value ^1,2,3^: comparison between normal pregnancies and preterm births, PTB or PPROM, respectively. PTB, spontaneous preterm birth; PPROM, preterm prelabor rupture of membranes; SLE, systemic lupus erythematosus; APS, antiphospholipid syndrome; RA, rheumatoid arthritis; SS, systemic sclerosis; DM, diabetes mellitus; AIT, autoimmune thyroiditis; ART, techniques of assisted reproduction; IVF, in vitro fertilization; ICSI, intracytoplasmic sperm injection; BMI, body mass index; PE, preeclampsia; FGR, fetal growth restriction; FMF, Fetal Medicine Foundation.

## Data Availability

The data presented in this study are available on request from the corresponding author. The data are not publicly available due to rights reserved by funding supporters.

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
