# Peer review of "Novel First-Trimester Prediction Model for Any Type of Preterm Birth Occurring before 37 Gestational Weeks in the Absence of Other Pregnancy-Related Complications Based on Cardiovascular Disease-Associated MicroRNAs and Basic Maternal Clinical Characteristics"

_biomedicines, 2022, doi:10.3390/biomedicines10102591_

Round 1
Reviewer 1 Report
This is an interesting paper that addresses the important clinical problem of determining the probability of preterm birth. I have several concerns about the manuscript.
1. The data show that adding an assessment of six critical miRNA's to your five selected clinical characteristics increases the predictability of preterm birth. However, a rigorous assessment of the predictability of preterm birth using ONLY the five clinical characteristics is needed. As assessment similar to the ones pictured in Figures 1-4, using ONLY the five clinical characteristics to show the added value of the miRNA's would be very helpful.
2. The column headings in Table 1 are confusing. "PTB" is an abbreviation for preterm birth. Please find a clearer way to distinguish between the cases designated as "Preterm birth" and "PTB".
3. Given that patients with hypertensive disorders of pregnancy and growth restriction were excluded from the study, did most of the patients with preterm birth or PROM have chorioamnionitis? Can you comment on the relationship between the six predictive miRNA's and inflammation?
4. The rate of preterm birth/PROM was very low, much lower than the world average, which is around 10%, and the patients were all Caucasian. In the United States, rates of preterm birth are much higher among African Americans than Caucasians. Can you comment on the generalizability of these data? Are the same six miRNAs likely to be predictive or preterm birth/PROM around the globe?
5. The English writing needs to be improved throughout the paper. Importantly, "destinated" should be changed to "destined"!
Reviewer 2 Report
I have followed the authors work with great interest and respect. I have several observations/criticisms.
1. My main concern for the present manuscript is missing information- is this a validation study of different subjects than those in their prior Discovery/Confirmation study cited in the current manuscript as reference 10? Or is it instead a re-analysis of the expression data from reference 10 combined with an expanded set of clinical markers? I suspect the answer is the latter of the two possibilities as the description of the subjects is seemingly identical in the two papers down to the most granular detail. A validation study would raise the impact, but an improved prognostic model using the original data in reference 10 would also be valuable to other investigators.
2. A comparison of the AUCs generated by the best prognostic models of each paper suggests little improvement was achieved by adding a larger set of clinical variables. This differs from preceding studies using other potential markers for preterm birth. Is this evidence that the miRNA markers selected for use are involved in early pathological events leading to PTB? Are these markers known to have intracellular activities that would seem part and parcel with a loss of myometrial quiescence?
3. There are several preceding publications that should be referenced. One is Cook et al in Scientific Reports 2019, and Weiner et al in 2021 BJOG. Both papers list a significant number of differentially expressed miRNAs for PTB. Is there any substantial overlap in the identified differentially expressed miRNAs among the three papers, and if not, why not?
4. The authors extract miRNA from whole blood where most predecessors focused on cell free plasma. How might the inclusion of maternal leukocytes and degranulated platelets distract attention from the organ of interest, the pregnancy?
5. Why in each of their analyses in reference 10 and the current manuscript do the authors exclude other pregnancy complications? Any test suitable for clinical deployment for 1st trimester screening will need to perform in the face of other pregnancy complications.
Reviewer 3 Report
Introduction
1. Please kindly add the aims and working hypothesis for this research.
Methods.
2. The sample size calculation and setting of the study is not provided.
3. The statistical method is unclear.
Discussion
4. Future work will involve the use of machine learning techniques for predicting preterm labor and measuring heart rate variability or electrohysterogram.
Round 2
Reviewer 1 Report
I believe that all of my concerns have been addressed and the manuscript is now much improved. I believe it is now publishable.
Author Response
Thank you very much for positive comments and recommendation for publication.
Reviewer 2 Report
The manuscript has clearly been improved and all queries answered by the authors.
There is one error in the authors' response. I believe the incorrectly quote Weiner et al. A reread of the Weiner manuscript indicates only Let 7g of the 13 differently expressed mIR was used for the associated validation study.
Author Response
We thank the reviewer for the notice of this issue. The correct information is that the Weiner manuscript indicates that only Let-7g of the 13 differently expressed microRNAs was used for the associated validation study.